# Analysis of Tianeptine in Dietary Supplements

**Jared T. Seale [1], Emily A. Garden [2], John M. T. French [3] and Owen M. McDougal [1,\*]**

[1] Department of Chemistry & Biochemistry, Boise State University, Boise, ID 83725, USA
[2] Department of Chemistry, Montana Technological University, Butte, MT 59701, USA
[3] School of Medicine, University of Missouri, Columbia, MO 65212, USA
\* Correspondence: owenmcdougal@boisestate.edu; Tel.:+1-208-426-3964

**Abstract:** In the United States (US), tianeptine is sold as a dietary supplement under the trade name Tianaa™. Tianeptine is a synthetic drug prescribed by physicians as an antidepressant in parts of Europe, Asia and South America. The drug is not permitted for use by physicians in the US, because it is a μ-opioid receptor agonist with a propensity for severe addiction. As the incidence of Tianaa™-related opioid addiction across the southern US escalates, the current study aimed to quantify tianeptine in over-the-counter Tianaa™ White, Red, and Green products. The results of this investigation measured tianeptine levels between 3.1 and 10.9 mg per 531 mg capsules. Tianaa™ White capsules consistently contained the least tianeptine, while Green had the most. The close inspection of Tianaa™ products showed that capsule mass varied by as much as 16% from label claim, and the amount of tianeptine per capsule varied by as much as 35% from the average measured amount for each product. Tianaa™ Red contains kava leaf extract, which led to the identification of four kavapyrone components by mass spectrometry. The data presented provide insight into tianeptine quantity and capsule mass variation for Tianaa™ supplements sold to customers naive to the risk of addiction.

**Keywords:** dietary supplement; μ-opioid-receptor agonist; addiction; antidepressant

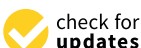



## 1. Introduction

Tianeptine is an antidepressant approved for prescription use in Europe, Asia, and Latin America [1]. Tianeptine is a synthetic tricyclic compound originally patented by Malen et al. at the French Society of Medical Research in 1973 [2]. The patent for the synthesis of tianeptine was issued to Biophore India Pharmaceuticals in 2010 [3]. The international patent describes the purpose of the drug as enhancing serotonin reuptake, but does not mention the specific working mechanism [3]. Tianeptine has a tricyclic structure, but it is classified as an atypical antidepressant because its mechanism of action is not consistent with other tricyclic antidepressants. Specifically, tianeptine aids in the regulation of plasticity in the amygdala, attenuation of stress-induced glutamate release, and reversal of stress-induced dystrophy of hippocampal CA3 dendrite [4–7]. With a dosage of 25–50 mg per day, tianeptine is as effective as classic antidepressants such as sertraline and fluoxetine [5]. The side effects of this drug are similar to selective serotonin reuptake inhibitors (SSRIs) in that it can cause headaches, dizziness, nausea, and abdominal pain [5].

Tianeptine has been shown to be a full μ-receptor opioid agonist that can lead to severe psychological or physical dependence. Experimentally, tianeptine appears to be approximately five times less potent than morphine [6]; its propensity for dependence is a characteristic of a United States Food and Drug Administration (FDA) Schedule II Controlled Substance commensurate, a list that includes morphine, methamphetamine, cocaine, fentanyl, phencyclidine (PCP) and others [6,8]. Mechanistically, as a full μ-opioid receptor agonist, tianeptine is thought to possibly cause increased dependence by stimulating the release of dopamine in the limbic system. This is carried out by the inhibition of GABAergic interneurons in the ventral tegmental area, leading to the disinhibition of dopaminergic

neurons [9]. Between 2011 and 2012, a multitude of fatal tianeptine overdoses influenced law makers in Turkey to classify tianeptine as a controlled substance and ban its use in the country [10]. A study from 2014 to 2017 by the Centers for Disease Control in the United States regarding incidents of tianeptine abuse revealed numerous concerning case studies of tianeptine overdose resulting in death [11,12]. By 2018, the reports of tianeptine addiction in psychiatric patients and incidents of overdose were prevalent in the literature [13,14]. The Drug Enforcement Agency (DEA) in the United States classifies tianeptine as not approved for medical use according to the FDA; however, it is not classified as a controlled substance in most states and is not regulated under the controlled substance act, despite μ-opioid receptor agonism and propensity for addiction [15]. At the time of this publication, only the states of Michigan, Alabama, Minnesota, Tennessee, and Oklahoma have classified tianeptine as a Schedule II drug [16]. Tianeptine is not approved for distribution in the United States as a prescription drug and does not meet the classification of a dietary supplement; however, it is sold in the over-the-counter supplement Tianaa™ [17].

There are three Tianaa™ products, differentiated by the names White, Red, and Green. The Tianaa™ products all contain tianeptine, and can be differentiated from one another by the addition of select organic material in their formulation. Tianaa$^{TM}$ White promotes "energy" and contains L-α-glycerylphosphorylcholine (α-GPC), a compound that promotes release of the neurotransmitter acetylcholine in the brain, and cytidine 5′-diphosphocholine (CDP choline), a molecule found naturally in the body that stabilizes cell membranes [18,19]. Tianaa™ Red is advertised to promote "rest and stress relief". This supplement has kava extract, which contains kavapyrones, used for their psychoactive effect to alleviate anxiety, but cautioned due to hepatotoxic side effects. Tianaa™ Green is marketed to induce "mellow moods", causing a sedating, calming, relaxing effect aimed at alleviating stress and anxiety.

The health risks posed by tianeptine usage in the United States have been recognized by several states, leading to the classification of the drug as a Schedule II substance. Tianeptine can be purchased online or over the counter in most states in the US within Tianaa™ products. Tianaa™ causes consumers to experience delirium, autonomic dysfunction, and increased dependence when taken in high doses. Severe withdrawal symptoms are most prevalent among frequent users of Tianaa™ products, whose dosage escalates with increased tolerance. Characteristic symptoms of withdrawal from Tianaa$^{TM}$ are agitation, insomnia, yawning, headache, restlessness, autonomic hyperactivity, and nausea. The principal reason for Tianaa™ usage is individuals seeking relief from chronic pain, which has not been relieved by surgical or pharmacologic interventions. Clinical studies correlating excessive consumption of tianeptine containing supplements such as Tianaa™ compared to clinical trials with controlled, prescribed tianeptine dosage of 25–50 mg demonstrate the propensity for dependence and risk of overdose associated with the supplements [20]. The current investigation targeted a quantitative understanding of tianeptine content across Tianaa™ products to gain insight into the alarming dependence profile exhibited by daily users seeking help from their addiction.

## 2. Materials and Methods

Tianaa™ product. Tianaa™ (Red, Green, and White) products were purchased online from MT Brands™ [21].

**Extraction of Tianeptine from Tianaa™ capsules**. One capsule from each class of Tianaa™ (Red, Green, and White) was divided equally into three samples and placed in centrifuge tubes. The samples were dissolved in three mL of 95% ethanol (Decon, UN1170), in which its solubility is 10 mg/mL [22], vortexed for 30 s, then centrifuged for five minutes at 4000 rpm. The superficial ethanol layer was extracted via pipet and then filtered into new centrifuge tubes. The three mL solutions were divided into two one-mL microcentrifuge tubes and placed in a centrifugal, reduced-pressure evaporator to remove the solvent (1 h). Added to each microcentrifuge tube was 250 μL of 100% ethanol (Spectrum Chemical, E1028), and the solution was then vortexed (4 min) to uniformly mix

it, and finally transferred to 2.0 mL autosampler vials for liquid chromatography—mass spectrometry (LC-MS) analysis.

**Tianeptine Identification and Quantification**. Tianeptine was identified and quantified in Tianaa™ samples using OpenLab CDS 2 software operating an Agilent 1260 Infinity II high-pressure liquid chromatography system coupled to a diode array detector and single quadrupole electrospray ionization detector (HPLC-DAD-MS), and commercially available standards. Samples were injected on a Waters Xterra MS $C_{18}$ column (5 μm, 2.1 × 150 mm) maintained at 40 °C at a volume of 5 μL. The mobile phase solvent system consisted of 0.1% trifluoroacetic acid (TFA) in water (solvent A) and 0.1% TFA in acetonitrile (solvent B) with a 0.5 mL/min flow rate. For Tianaa™ analysis, the gradient elution started at 15% solvent B and increased to 50% solvent B over 10 min. After 10 min, the solvent composition was returned to 15% solvent B between minutes 10 and 15. The electrospray ionization source was operated under the following conditions: positive ion-mode, 4000 to −4000 V between capillaries, 7.0 L/min $N_2$ drying gas at 300 °C and nebulizer pressure of 15 psi. Tianeptine was quantified using standard curve using a tianeptine sodium salt hydrate (>98% purity, CAS RN 30123-17-2) purchased from TCI America diode array detector (DAD) measuring absorbance at 220 nm.

**Statistical Methods of Analysis**. Statistical values for the amount of tianeptine present per capsule for each Tianaa™ product were calculated using Microsoft Excel. An unpaired two-tailed *t*-test was used to determine *p*-value.

## 3. Results

### 3.1. Tianaa™ Product Comparison—Red, Green, and White

#### 3.1.1. Tianeptine and Kavapyrone Analysis

To identify and quantify tianeptine in each type of Tianaa™, a commercially available tianeptine standard was used as a direct comparison for Tianaa™ sample extracts. Tianeptine was identified in each sample with a retention time between 7.616 and 7.716 min and $[M + H]^+$ of 437.2 *m/z* (Figure 1). Although tianeptine was the focus of this study, both chromatographic and mass spectral data identified the presence of the kava leaf compounds dihydromethysticin, kavain, desmethoxyyangonin, and yangonin, within the ethanolic extract of Tianaa™ Red (Figure 1A and Table 1). The identities of these compounds were consistent with results from previously published work through the observation of expected $[M + H]^+$ molecular ions [23]. The identified compounds are constituents of kava leaf extract and are added to Tianaa™ Red to reduce anxiety. Tianeptine concentration across Tianaa™ products was determined using standard curve ($R^2$ = 0.9999), where the limit of detection and limit of quantification for tianeptine was 0.51 mg/L and 1.55 mg/L, respectively. On average, the mass of tianeptine per capsule for Tianaa™ Red, Tianaa™ Green, and Tianaa™ White was 8.06 ± 0.03 mg, 8.08 ± 0.03 mg and 4.55 ± 0.04 mg, respectively (Figure 2). The percent deviation of the mass of tianeptine from the mean for each type of Tianaa™ ranged from −33–35%, −32–36%, and −26–30% for Tianaa™ Red, White, and Green, respectively.

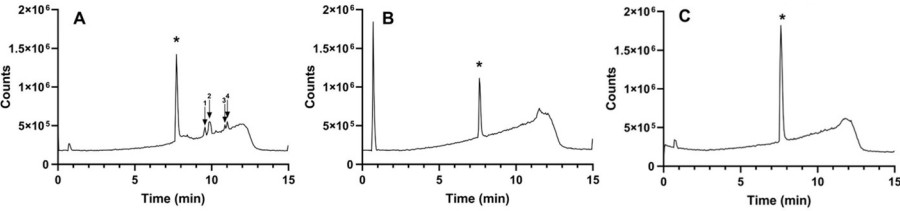

**Figure 1.** Total ion chromatograms (TIC) for (**A**) Tianaa™ Red, (**B**) Tianaa™ White, and (**C**) Tianaa™ Green. Tianeptine (*) was identified with a retention time of 7.616–7.716 min. The proposed identity of numbered components in panel (**A**) are listed in Table 1.

**Table 1.** Kava leaf extract: compounds predicted in the Tianaa™ Red ethanolic extract.

| Compound Name | Structure | Retention Time (min) | $[M + H]^+$ (*m/z*) |
|---|---|---|---|
| Dihydromethysticin (1) | | 9.570 | 277.1 |
| Kavain (2) | | 9.820 | 231.1 |
| Desmethoxyyangonin (3) | | 10.872 | 229.1 |
| Yangonin (4) | | 11.022 | 259.1 |

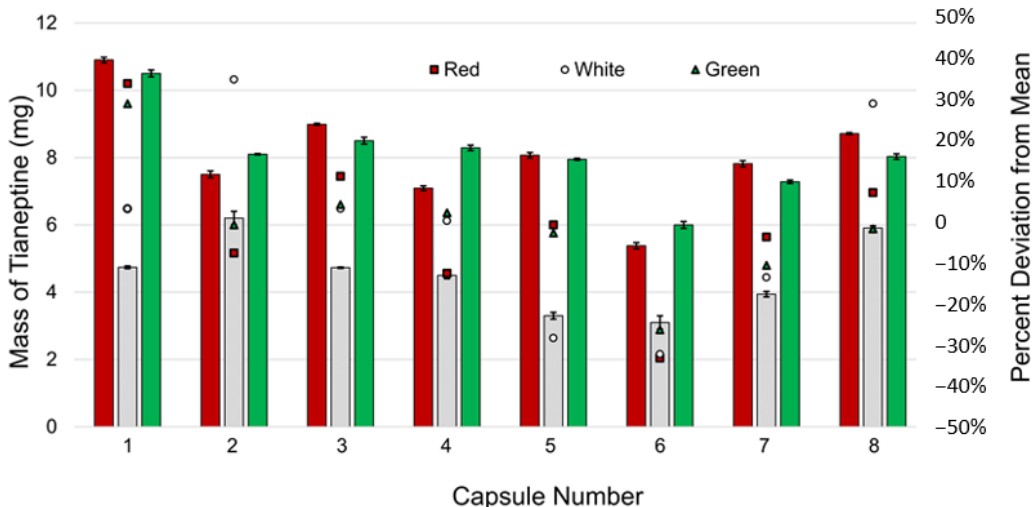

**Figure 2.** Mass of tianeptine within eight capsules of Tianaa™ compared to the percent deviation of tianeptine from the observed mean for each type of Tianaa™. From left to right, the bars within this graph represent Tianaa™ Red, White, and Green. Markers represent the percent deviation from the mean of tianeptine for that capsule.

### 3.1.2. Capsule-to-Capsule Comparison

Following the identification of tianeptine in Tianaa™, a capsule-to-capsule comparison was performed to investigate quantity variation of tianeptine within each Tianaa™ product. The total capsule mass ranged between 445 and 601 mg, whereas the reported capsule mass from the manufacturer is 531 mg (Figure 3). The percent mass of tianeptine compared to the total mass of the capsule ranged between 0.6 and 2.4%; however, the positive and negative deviations of tianeptine mass from the averaged mass were as high as 36% and 33%, respectively. Altogether, these data suggest that the mass of tianeptine

within each capsule was inconsistent within and between each type of Tianaa^TM, and not dependent upon the total mass of the capsule substrate.

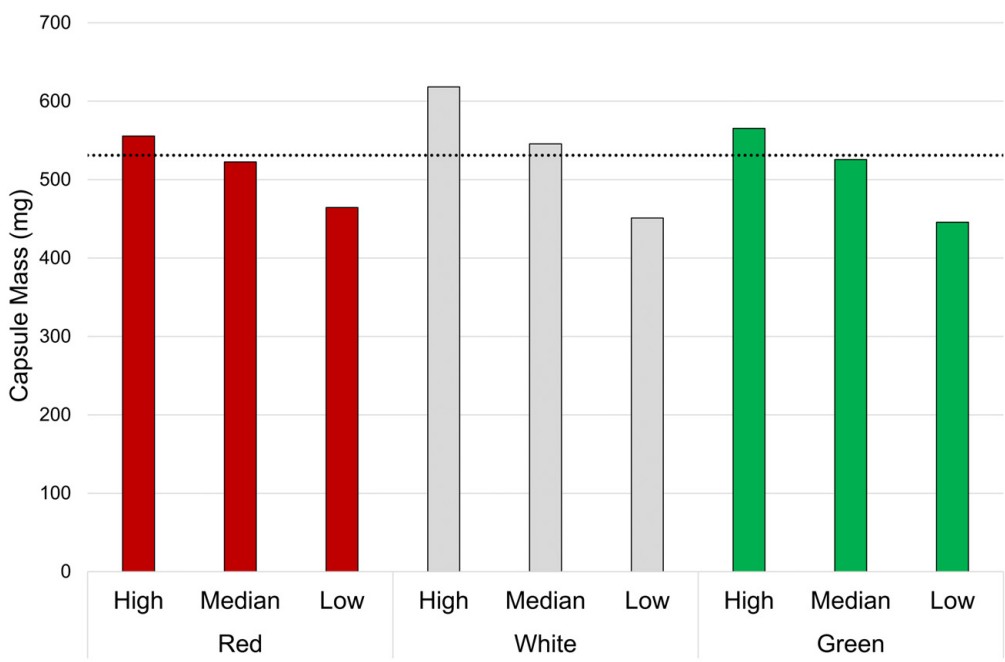

**Figure 3.** Observed high, median, and low capsule mass for Tianaa^TM Red, White, and Green compared to the expected mass of 531 mg (marked by a dashed line).

## 4. Discussion

The Food and Drug Administration has not approved Tianna^TM for consumption. According to the Tianna^TM product label, individuals that are 18 years of age or older can take two capsules daily as needed, but not to exceed three capsules in a 24 h period. Product warnings are listed that include consumer age at or above 18 years, recommendation not to take if pregnant, nursing, or using monoamine oxidase inhibitors, driving or operating machinery, precaution due to potential for addiction. There is no discussion on the product label associated with tianeptine content or direct risk associated specifically to tianeptine. For this reason, the current study assessed the mass of tianeptine present in capsules for each class of Tianaa^TM, which were found to deviate between 0 and 36% from their respective average values. From the analysis of eight capsules from each Tianaa^TM class where the tianeptine content in each capsule was measured in triplicate, it was determined that tianeptine mass in Tianaa^TM White was statistically ($p$-value $< 0.01$) different from that found in Tianaa^TM Red or Tianaa^TM Green. This difference in tianeptine content was not due to differences in the total mass of each capsule. Between 0.6 and 2.4% of a capsule's mass was tianeptine (see Table 2). The calculated percent variance of tianeptine was based on an average, because the manufacturer does not report the mass of tianeptine per capsule. The difference in mass between individual capsules and an average capsule mass was consistently greater than 10%, with positive and negative deviations of 36% and 33%, respectively. The calculated capsule masses had low standard deviations (0.02–0.2 mg), suggesting that the variation in tianeptine mass is likely due to the manufacturing process.

**Table 2.** Percent mass of tianeptine present in whole capsule.

| Capsule | Red | Green | White |
|---|---|---|---|
| | % Mass | % Mass | % Mass |
| 1 | 2.35 | 1.05 | 2.36 |
| 2 | 1.36 | 1.04 | 1.44 |
| 3 | 1.81 | 0.85 | 1.67 |
| 4 | 1.39 | 0.82 | 1.50 |
| 5 | 1.47 | 0.59 | 1.55 |
| 6 | 1.07 | 0.66 | 1.06 |
| 7 | 1.51 | 0.69 | 1.40 |
| 8 | 1.66 | 1.00 | 1.53 |

The mass of Tianaa™ capsules varied between and within Tianaa™ products. In the case of Tianaa™ White, measurement of mass across three capsules varied by ≥15% compared to the reported value on the bottle, which exceeds the United States GMP standards [18]. Tianaa™ Red and Green capsules varied in mass between 5 and 16%, where one capsule exceeded the ≥15% benchmark. The US GMP standards report that no more than two capsules may exceed 15% if the capsules are over 300 mg in mass; the bottle label for Tianaa™ products lists the capsule mass to be approximately 531 mg [24].

The results of the current study suggest that Tianaa™ brand tianeptine supplements do not conform with United States GMP standards. The opioid activity of tianeptine and observed variability in drug content and capsule mass suggest that enhanced regulatory oversight is warranted. In countries where tianeptine is legal for medical use, a prescribed daily dose is 13.5–50 mg per day. Based on our analysis of Tianaa™ Red and Green products, a median tianeptine level of approximately 8.06 ± 0.03 mg per capsule will deliver a therapeutic dose of the drug in two to six capsules [25]. Lauhan et al. described a compilation of studies investigating tianeptine dependence or abuse where the average daily dose of tianeptine was on the order of 1924 mg across 65 patients, further solidifying a basis for risk of tianeptine addiction due to overconsumption of Tianaa™ [20]. Further investigation of the quantities of tianeptine in supplements may assist in informing clinicians of patient-regulated tianeptine use.

The importance of this research comes back to what is seen being seen in medical facilities. Clinically, Tianna™ has been correlated with severe withdrawal symptoms [20]. The increased tolerance associated with Tianna™ and increased risk of dependence from the supplement is most likely due to a multitude of effects induced by tianeptine on receptors, including glutaminergic, dopaminergic, and GABA, in various neuronal tracts in the brain. Additionally, other natural products in the formulation of Tianaa™ supplements may attenuate the effects on neural circuits. This study draws attention to the risk of unregulated tianeptine-containing supplements and the variation in active components that may exist within and between capsules of the same or related products.

## 5. Conclusions

An analysis of Tianaa™ products reveals inconsistent tianeptine content and capsule mass, which exceed good manufacturing practice standards. The product variation for this nutraceutical supplement, coupled with concerning addiction potential for tianeptine, are expected to lead to inevitable increases in incidents of overdose and addition. The long-term repercussions of tianeptine addiction include increased tolerance to Tianaa™, which requires more capsules for daily consumption, and severe withdrawal symptoms that include intensive agitation, nausea, headache, and autonomic hyperactivity [20]. Patients requiring medical intervention for tianeptine abuse are treated with medication regimens that include the administration of methadone, buprenorphine, or clonidine. To justify labeling Tianaa™ products as dietary supplements, nutrients like kava extract, *Tribulus terrestris* fruit, niacin, and others are added, but masking tianeptine is dangerous for consumers that are not aware of the risks of overdose and addiction.

**Author Contributions:** Conceptualization, J.T.S., J.M.T.F. and O.M.M.; methodology, J.T.S. and E.A.G.; validation, J.T.S., E.A.G., J.M.T.F. and O.M.M.; formal analysis, J.T.S., E.A.G., J.M.T.F. and O.M.M.; investigation, J.T.S. and E.A.G.; resources, O.M.M.; data curation, O.M.M.; writing—original draft preparation, J.T.S. and E.A.G.; writing—review and editing, J.M.T.F. and O.M.M.; visualization, J.T.S. and E.A.G.; supervision, O.M.M.; project administration, J.M.T.F. and O.M.M.; funding acquisition, O.M.M. All authors have read and agreed to the published version of the manuscript.

**Funding:** This research and the APC was funded by the Institutional Development Awards (IDeA) from the National Institute of General Medical Sciences of the National Institutes of Health under Grants #P20GM103408, P20GM109095, and 1C06RR020533. We also acknowledge support from The Biomolecular Research Center at Boise State University (BSU) Biomolecular Research Center, RRID:SCR_019174, with funding from the National Science Foundation, Grants #0619793 and #0923535; the M. J. Murdock Charitable Trust; Lori and Duane Stueckle, and the Idaho State Board of Education.

**Institutional Review Board Statement:** Not applicable.

**Informed Consent Statement:** Not applicable.

**Data Availability Statement:** Data supporting reported results can be requested from the corresponding author at owenmcdougal@boisestate.edu.

**Conflicts of Interest:** The authors declare no conflict of interest.

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
