# Peer review of "Analysis of Tianeptine in Dietary Supplements"

_nutraceuticals, doi:10.3390/nutraceuticals3030034_

Round 1
Reviewer 1 Report (Previous Reviewer 3)
Comments and Suggestions for Authors
The main shortcomings of the work have been corrected. In my opinion, the following changes are still needed:
1. Supplementing the materials and methods section with a description of the statistical methods used (their results are given in the Discussion section)
2. Presenting at least some of the obtained results in the form of a graph.
3. Discussion of the number of daily capsules recommended by the manufacturer in the context of the determined tianeptine content and the risk it entails.
Author Response
Reviewer 1
The main shortcomings of the work have been corrected. In my opinion, the following changes are still needed:
Comment 1. Supplementing the materials and methods section with a description of the statistical methods used (their results are given in the Discussion section)
Response: The following section was added in the Materials and Methods section of the paper on page 3 to address this comment.
Statistical Methods of Analysis.
Statistical values for the amount of tianeptine present per capsule for each Tianaa™ product were calculated using Microsoft Excel. An unpaired two-tailed t-test was used to determine p-value.
Comment 2. Presenting at least some of the obtained results in the form of a graph.
Response: In an effort to clean up the presentation of information, Figure 1 has been modified to omit panel D due to resolution challenges. Table 2 results have been represented in bar graph form as Figure 2. This figure shows the average mass of tianeptine (mg) over eight capsules, each capsule assessed in triplicate, and the percent deviation from the mean value is also displayed. Similarly, Table 3 content has been represented in bar graph form as Figure 3. This figure shows a dotted line to represent the mass of capsule reported on the product label at 531 mg, and the observed masses of the capsules at high, median, and low measurement. This figure provides context for the variation observed within and across product types for Tianaa™ products.
Comment 3. Discussion of the number of daily capsules recommended by the manufacturer in the context of the determined tianeptine content and the risk it entails.
Response: To address this comment the following content was added to begin the discussion section of the paper on page 5. “The Food and Drug Administration has not approved TiannaTM for consumption. According to the TiannaTM product label, individuals that are 18 years of age or older can take two capsules daily as needed, but not to exceed three capsules in a 24 hr period. Product warnings are listed that include consumer age at or above 18 years, recommendation not to take if pregnant, nursing, or using monoamine oxidase inhibitors, driving or operating machinery, precaution due to potential for addiction. There is no discussion on the product label associated with tianeptine content or direct risk associated specifically to tianeptine. For this reason, the current study assessed the mass of tianeptine present in capsules for each class of TianaaTM, which were found to deviate between 0-36% from their respective average values.”
Reviewer 2 Report (New Reviewer)
Comments and Suggestions for Authors
One major limitation is that reproduction of experiment is limited because the commercial sources or number of commercial sources of TianaaTM is not listed in the material methods section.
Table 1 is misleading because these “predicted” compounds were not actually quantified. Unless table 1 can be justified for another reason it is suggested that it should be removed as to mislead the reader.
There is some concern that the authors are overstating the risks associated with TiannaTM and tianeptine. Last paragraph (lines 72-84) of the Introduction many of the adverse effects of tianeptine are noted by there are not references listed. Likewise, there are no references for last paragraph of the Discussion (lines 189-197) and stated consumer adverse interactions in the Conclusion (lines 200-208).
The authors should clearly state the accepted product variation accepted by United States GMP standards. Based on what is written “the US GMP standards report that no more than two capsules may exceed 15% if the capsules are over 300 mg in mass. The bottle label for Tianaa™ products lists the capsule mass to be approximately 531 mg (Lines 173-175). This suggests that the variation in mass is acceptable, Moreover, it should be stated there was only 1 capsule in Green and 1 capsule in White that were 16%. Again, the commercial sources of these capsules were not identified.
Author Response
Reviewer 2
Comment 1. One major limitation is that reproduction of experiment is limited because the commercial sources or number of commercial sources of TianaaTM is not listed in the material methods section.
Response: Tianaa™ is a proprietary blend of natural herbs and nootropics. The study focused on the amount of tianeptine present within the proprietary blend that is not listed on the label nor conveyed to the consumer. The distributor of Tianaa™ has been included in the materials and methods section of the paper on page 2 using the following added text.
Tianaa™ product. Tianaa™ (Red, Green, and White) products were purchased online from MT Brands™ [21].
Comment 2. Table 1 is misleading because these “predicted” compounds were not actually quantified. Unless table 1 can be justified for another reason it is suggested that it should be removed as to mislead the reader.
Response: The description of the content in Table 1 was incorrect prior. The wording has been corrected to now read as follows.
Although tianeptine was the focus of this study, both chromatographic and mass spectral data identified the presence of the kava leaf compounds dihydromethysticin, kavain, desmethoxyyangonin, and yangonin, within the ethanolic extract of TianaaTM Red (Figure 1A and Table 1). The identities of these compounds were consistent with results from previously published work through the observation of expected [M+H]+ molecular ions [21]. The identified compounds are constituents of kava leaf extract and are added to TianaaTM Red to reduce anxiety.
Comment 3. There is some concern that the authors are overstating the risks associated with TiannaTM and tianeptine. Last paragraph (lines 72-84) of the Introduction many of the adverse effects of tianeptine are noted by there are not references listed. Likewise, there are no references for last paragraph of the Discussion (lines 189-197) and stated consumer adverse interactions in the Conclusion (lines 200-208).
Response: The following citation has been added at the three places identified by the reviewer to draw the readers attention to case reports associated with tianeptine abuse and dependence.
- Lauhan, R.; Hsu, A.; Alam, A.; Beizai, K. Tianeptine Abuse and Dependence: Case Report and Literature Review, Psychosomatics. 2018, 59(6), 547-553, https://doi.org/10.1016/j.psym.2018.07.006.
Comment 4. The authors should clearly state the accepted product variation accepted by United States GMP standards. Based on what is written “the US GMP standards report that no more than two capsules may exceed 15% if the capsules are over 300 mg in mass. The bottle label for Tianaa™ products lists the capsule mass to be approximately 531 mg (Lines 173-175). This suggests that the variation in mass is acceptable, Moreover, it should be stated there was only 1 capsule in Green and 1 capsule in White that were 16%. Again, the commercial sources of these capsules were not identified.
Response: The supplier of Tianaa™ has been included and cited as reference 21. Tables 2 and 3 have been replaced with bar graphs in an effort to more clearly show variation within product type and across product types for tianeptine content (Figure 2) and capsule mass (Figure 3), respectively. The following content has been edited to address this comment.
Tianaa™ Red and Green capsules varied in mass between 5-16%, where one capsule exceeded the ≥15% benchmark.
Round 2
Reviewer 2 Report (New Reviewer)
Comments and Suggestions for Authors
The authors have addressed all of my previous concerns.
This manuscript is a resubmission of an earlier submission. The following is a list of the peer review reports and author responses from that submission.
Round 1
Reviewer 1 Report
Comments and Suggestions for Authors
In this manuscript, the authors contribute with a communication based on the analysis of the quantity of tianeptine in OTC Tianaa™ White, Red, and Green products. The presentation of communication is quite clear, easy to follow, and informative. Abstract is reasonable, and text is focused on the topic.
This communication is especially interesting because tianeptine is a drug authorized in different countries as an antidepressant, but it has different mechanisms of action; one of them as a μ-opioid-receptor agonist, which determines the possibility of inducing an addictive disorder. The results are very relevant because it is a marketed product that does not meet the United States good manufacturing practice standards.
I allow myself to make a series of moderate and minor specific comments:
1) The communication could benefit if authors, in the INTRODUCTION, will better explain the different pharmacodynamic properties of tianeptine: it increases the reuptake of serotonin, therefore its extracellular rate decreases; its chronic administration does not alter the density or affinity of more than a hundred classic receptors related to depression; its action on mu opioid receptors could explain the release of dopamine in the limbic system and its participation in the modulation of glutamatergic mechanisms; etc.
2) It would be convenient to indicate that the potency of tianeptine on mu receptors is 6 times less than that of morphine.
3) Although tianeptine has a tricyclic structure, technically it cannot be classified as a tricyclic antidepressant, but rather as an atypical antidepressant.
4) As the authors indicate, one of the limitations of the study is the small sample of capsules analyzed, making it necessary to provide a greater number of samples to confirm the results of the study.
5) The authors would discuss better the relationship between the mean amount of tianeptine observed in the samples with the daily dose when prescribed as an antidepressant (13.5-50 mg per day, versus 8.06 ± 0.03 mg per capsule of TiannaTM). This would very well explain the addiction risks of excessive consumption of TiannaTM.
Some minor typographical corrections:
1) The authors must standardize the way of writing some words, such as tianeptine or midwest, sometimes with a capital letter and other times with a lower case letter.
2) The authors should pay attention to certain typographical errors, such as separating some words that appear right after the % symbol (15%and; 15%compared; etc.).
3) The authors must review the citations in the REFERENCES section: adjust well the number of authors that are cited in each reference (sometimes 3 et al. and in others more than 6); in citations 18 and 19 the abbreviated name of the journal appears; etc.
Reviewer 2 Report
Comments and Suggestions for Authors
Reviewer 3 Report
Comments and Suggestions for Authors
Although the manuscript deals with an important issue, it is, in my opinion, poorly prepared.
The introduction part lacks information on what, according to the manufacturer, the individual varieties of Tianna dietary supplements contain (are they different plant extracts depending on the type?).
The part concerning the method of determination of tianeptine in the serum is completely incomprehensible - have such determinations been performed? Were any patients/volunteers involved in the study?
The developed extraction method does not include recovery.
The authors use the term statistically significant differences, on what basis? There is no mention of statistical analysis in the entire manuscript.
Tianeptine and kava qualitative and quantitative analysis - kava is a plant can you quantify a plant?
The "results" section begins with an excerpt from the manuscript template.
The study is poorly planned and poorly described.